# Prediction of Immunotherapy Response in Hepatocellular Carcinoma Patients Using Pretreatment CT Images

**DOI:** 10.3390/diagnostics15162090

**Published:** 2025-08-20

**Authors:** Ji Hye Min, Pin-Jung Chen, Touseef Ahmad Qureshi, Sehrish Javed, Yibin Xie, Linda Azab, Lixia Wang, Hyun-seok Kim, Debiao Li, Ju Dong Yang

**Affiliations:** 1Karsh Division of Gastroenterology and Hepatology, Cedars-Sinai Medical Center, Los Angeles, CA 90048, USA; minjh1123@gmail.com (J.H.M.); hyunseok.kim@cshs.org (H.-s.K.); 2Department of Radiology, Samsung Medical Center, Sungkyunkwan University School of Medicine, Seoul 06351, Republic of Korea; 3Samuel Oschin Comprehensive Cancer Institute, Cedars-Sinai Medical Center, Los Angeles, CA 90048, USA; pin-jung.chen@cshs.org; 4Biomedical Imaging Research Institute, Cedars-Sinai Medical Center, Los Angeles, CA 90048, USA; touseefahmad.qureshi@cshs.org (T.A.Q.); sehrish.javed@cshs.org (S.J.); yibin.xie@cshs.org (Y.X.); linda.azab@cshs.org (L.A.); lixia.wang@cshs.org (L.W.); 5Department of Bioengineering, University of California, Los Angeles, CA 90095, USA; 6Comprehensive Transplant Center, Cedars-Sinai Medical Center, Los Angeles, CA 90048, USA

**Keywords:** hepatocellular carcinoma, immunotherapy, response, radiomics, machine learning

## Abstract

**Background/Aims:** Predicting treatment response to immunotherapy in hepatocellular carcinoma (HCC) is essential to improve clinical outcomes with personalized treatment strategies. This study aims to develop an AI-driven prediction model using radiomic analysis from the liver and viable HCCs on pretreatment CT to differentiate responders from non-responders. **Methods:** HCC patients who received immunotherapy between 2016 and 2023 with pretreatment CT scans were included. Radiomic features were extracted from the whole liver and the viable HCCs on the portal venous phase CT prior to immunotherapy. Multiple machine learning models were trained for binary classification to predict treatment response, initially using liver features (Model 1), and subsequently including both liver and tumor features (Model 2). Model performance was evaluated using three-fold cross-validation. **Results:** Among 55 HCC patients (median age, 69; 76.4% male) who received immunotherapy, 21 (38.2%) were responders and 34 (61.8%) non-responders by mRECIST criteria. Over 5000 radiomic features were extracted from pretreatment CT scans of the liver and viable tumors, of which approximately 100 were predictive of treatment response. Model 1 (liver) achieved an average accuracy of 77%, sensitivity of 76%, and specificity of 78%. Model 2 (liver and tumor) demonstrated improved performance, with accuracy, sensitivity, and specificity of 86%, 70%, and 94%, respectively, supporting the value of combined liver–tumor radiomics in treatment response prediction. **Conclusions:** This pilot study developed an AI-based model using CT-derived radiomic features to predict immunotherapy response in HCC patients. The approach may offer a non-invasive strategy to support personalized treatment planning using pretreatment CT scans.

## 1. Introduction

Hepatocellular carcinoma (HCC) is the most common primary liver cancer and a major cause of cancer-related death [1,2]. In 2020, approximately 830,200 deaths were attributed to liver cancer, with HCC constituting the majority [1]. Despite improvements in treatment modalities, the 5-year survival rate for HCC remains dismal at 22%, primarily due to late-stage diagnoses. Recently, immunotherapy has transformed treatment paradigms and improved prognosis in advanced HCC [3]. However, the success of immunotherapy varies among patients, and accurate prediction of therapeutic response remains a major clinical challenge [4].

Accurate pretreatment prediction of therapeutic response is of utmost importance for optimizing clinical decision-making in HCC management. The heterogeneity of HCC, both in terms of tumor biology and its response to treatment, complicates this process. Tumors may exhibit different responses to various therapeutic modalities. Accurately identifying which patients are more likely to respond to a specific treatment could lead to better personalized care, helping clinicians select the most appropriate and effective treatment plans for individual patients.

Imaging modalities such as computed tomography (CT) and magnetic resonance imaging (MRI) offer valuable information about tumor size, number, morphology, location, and vascular involvement. However, they are not reliable for predictions of treatment outcomes. Furthermore, the molecular and genetic heterogeneity of HCC tumors complicates the prediction of treatment efficacy based on imaging alone [5]. Artificial intelligence (AI) and machine learning offer promising solutions for predicting treatment outcomes in HCC by identifying imaging biomarkers beyond the scope of conventional assessment. Machine learning models extract radiomic features—such as texture, shape, and heterogeneity—from pretreatment CT scans, enabling the quantification of complex tumor phenotypes that can influence treatment response, even beyond what is visible through conventional imaging [6,7,8]. Recent studies have demonstrated that these models outperform traditional criteria, such as Barcelona Clinic Liver Cancer (BCLC) staging, in predicting outcomes of locoregional and systemic therapies [9,10,11,12]. These models can provide a more comprehensive understanding of tumor biology, enhancing clinicians’ ability to make informed decisions about which therapies are most likely to succeed.

This study aims to develop a predictive model for immunotherapy treatment outcomes in HCC patients based on pretreatment CT scans. We focused on radiomic feature extraction from the liver and viable HCC to classify patients as responders or non-responders to immunotherapy.

## 2. Materials and Methods

### 2.1. Data Collection and Preprocessing

The study protocol was reviewed and approved by the Institutional Review Board of Cedars-Sinai Medical Center (IRB No: STUDY00000341; approval date: 11 November 2019) and conducted in accordance with both the Declarations of Helsinki and Istanbul. Patient consent was waived due to the retrospective nature of the study. Clinical and imaging data were retrospectively collected from institutional databases through a review of medical records. We identified 148 consecutive HCC patients who received immunotherapy treatment (pembrolizumab, nivolumab, atezolizumab, durvalumab, and tremelimumab) between June 2016 and October 2023. The inclusion criteria were as follows: (a) patients with diagnosis of HCC (either imaging or pathological diagnosis), (b) patients with availability of a triphasic CT scan within 3 months of immunotherapy initiation, and (c) patients who had at least 1 follow-up with treatment response assessment after at least 2 doses of immunotherapy treatment. The exclusion criteria were as follows: (a) no available triphasic CT scan within 3 months before immunotherapy (*n* = 62), (b) inappropriate CT image quality (*n* = 1), and (c) unavailable for treatment response assessment (*n* = 30) (Appendix A).

Clinical characteristics, including age, sex, etiology of chronic liver disease, BCLC staging, and treatment details, were collected from electronic medical records. These clinical variables were not used for model training or feature selection, but only to describe the study population, as shown in the workflow (Figure 1). Tumor response following the initiation of immunotherapy was assessed every 8–12 weeks using contrast-enhanced CT or MRI, in accordance with the modified Response Evaluation Criteria in Solid Tumors (mRECIST) [13]. Patients showing a complete response (CR) or partial response (PR) were classified as responders, while those with stable disease (SD) or progressive disease (PD) were considered non-responders.

### 2.2. CT Acquisition and Preprocessing

Abdominal CT examinations were conducted prior to initiation of immunotherapy. For the portal venous phase, scanning began 70 s after the start of contrast injection. All pretreatment CT scans were acquired with sufficient quality, featuring a slice resolution of 512 × 512. The voxel values in the scans were normalized using min-max scaling to a range between 0 and 1. No smoothing, interpolation, or additional preprocessing was applied to the original voxel data.

### 2.3. Manual Segmentation

Two experienced radiologists (with more than ten years of experience in liver imaging) independently outlined the whole liver and viable HCCs in each slice. Segmentations were performed on the portal venous phase images. After an initial independent segmentation, the radiologists achieved 87% consistency in their labeling outcomes. The two readers then shared their annotations and reassessed their initial labels, resulting in 96% agreement. Discrepancies (4%) were resolved through collaborative discussion, leading to a final consensus. Labeling was performed using 3D Slicer (version 5.6.1) and ITK-Snap software (version 4.0.2), and the segmented data were exported in DICOM format [14]. Additionally, any information from the radiology reports was excluded to eliminate potential labeling biases.

### 2.4. Treatment Outcome Prediction Modelling

A machine learning model was developed using comprehensive radiomic analysis of both the liver and viable HCCs from pretreatment CT scans to automatically classify patients as responders or non-responders. The methodology was developed and evaluated using a three-fold cross-validation. In each of the three folds, the classifier used a unique subset (two-thirds) of the entire data for radiomic analysis and classifier training, and one-third for classifier testing. Specifically, scans of 14 responders and 23 non-responders were used for radiomic analysis and classifier training, and 7 responders and 11 non-responders’ scans were used for classifier testing. Figure 1 shows an illustration of the workflow of the radiomic model development. Representative axial CT images from each response group (responder and non-responder) are provided in Appendix A, including manual segmentation of the liver and viable tumor.

### 2.5. Radiomic Analysis and AI-Driven Model Development

#### 2.5.1. Extraction of Radiomic Features

Radiomic features were extracted from pretreatment CT scans using manually segmented regions of interest (ROIs), encompassing the entire liver parenchyma and the viable HCC. For each patient, two radiomic feature sets were generated—one from the liver and the other from the viable tumor. Feature extraction was performed using PyRadiomics, which complies with the Imaging Biomarker Standardization Initiative (IBSI). All radiomic metrics were quantified as a single numerical output based on predefined mathematical definitions. For example, liver attenuation was quantified as the mean gray-level intensity of all voxels within the 3D segmented liver volume.

A core group of over 100 base features, including intensity-, texture-, shape-, and size-based metrics was computed from each ROI. These encompassed first-order statistics (e.g., kurtosis, percentiles), texture features derived from gray-level co-occurrence matrix (GLCM), run-length matrix (GLRLM), size zone matrix (GLSZM), and dependence matrix (GLDM), as well as shape-based descriptors (e.g., volume, surface area, sphericity) (Table 1).

Derived features were generated by using different combinations of three key radiomic parameters: bin size (21 to 28), kernel size (1–5), and angle. The discretization reduces noise by converting the continuous values of voxels into discrete counterparts, avoiding contrast variation among CT images from different scanners. For example, a voxel value of 230 is not meaningfully different from 231, as such minimal variations are likely due to noise rather than reflecting true spatial heterogeneity.

In total, approximately 5000 radiomic features were generated by applying all valid parameter combinations to the base features.

#### 2.5.2. Identification of Predictive Features

A comprehensive analysis of the extracted radiomic features was performed to identify those that significantly differed between responders and non-responders, both in the whole liver and in viable HCCs. Various analytical tools were employed, and the extracted features were subjected to several statistical tests, including the following:

*Data Consistency:* A set of clean and uniform features was obtained by removing those with zero, undefined, or null values in each set, ensuring consistency in the type and number of features.

*Significant Feature Identification:* Statistically significant features that differed between the two sets of scans for both the whole liver and the viable HCC were identified. Statistical methods including Student’s *t*-test and the Bhattacharya coefficient were applied independently to each of the extracted features to determine which ones significantly differed between the two comparisons.

*Feature Redundancy Reduction:* A significant feature may be identified with multiple versions, each with a unique combination of radiomic parameters. For instance, a feature X between responders and non-responders’ viable tumor might be significant in different combinations of bin size, kernel size, and angle. In such a case, only one version of the feature was randomly selected, and the rest were excluded to reduce feature redundancy.

*Cross-sets Feature analysis:* Multivariate analysis was performed using significant features to identify those that presented either direct or inverse correlation between the corresponding features extracted from the liver and viable HCC.

#### 2.5.3. Binary Classification of Responders and Non-Responders

To build a predictive model of treatment response, we trained five commonly used supervised machine learning classifiers, comprising the Naïve Bayes (NB), k-nearest neighbor (KNN), support vector machine (SVM), linear regression (LR), and decision tree (DT). These classifiers are considered fundamental tools in biomedical informatics due to their ease of implementation, relatively low computational demands, and capacity for interpretable outputs—attributes particularly valuable in early-stage, proof-of-concept research. The NB classifier applies Bayes theorem under the “naïve” assumption of conditional independence among features (i.e., assuming no interdependence between individual radiomic features, whether from the same or different anatomical regions). The KNN classifies assign class labels based on the distance to nearby labeled samples within the feature space. The SVM constructs a decision boundary that maximizes the margin between the radiomic feature representations of the two classes. The LR constructs a linear decision boundary between two classes by applying a logistic sigmoid function to the weighted combination of input features to avoid outliers. The DT classifier works by specifying nodes, each representing a radiomic feature: branches, each corresponding to a decision outcome of the test; and terminal nodes, each indicating one of the classification outcomes. These classifiers were selected due to their established utility in binary classification tasks, particularly in radiomics studies involving tumor characterization across various cancers, including those for colon cancer [16], breast cancer [17], and detection of early stage of pancreatic cancer [18]. Moreover, given the limited sample size and exploratory scope of this study, we deliberately avoided more complex or highly parameterized algorithms to reduce the risk of overfitting.

The prediction of treatment outcome of HCC was framed as a binary classification: responders (CR + PR) and non-responders (SD + PD). Each classifier was performed using two separate feature sets: first, using radiomic features derived from the whole liver alone; and second, using a combination of features from both the whole liver and viable HCC to assess the added value of liver information and identify the optimal combination that maximized classification accuracy. A systematic approach ensured that each classifier selected a minimal set of significant features to avoid overfitting, enhancing generalizability without compromising performance. The Recursive Feature Elimination (RFE) method was employed with five classifiers to eliminate weaker features by comparing the overall training accuracies achieved with different feature combinations. Each classifier was limited to selecting a maximum of 10 features for each of the whole liver and viable HCC. The final model was designed to automatically classify each pretreatment abdominal CT scan as either a responder or a non-responder.

## 3. Results

### 3.1. Clinical Characteristics

A total of 55 patients were included, with a median age of 69 (IQR 63–73) years, and 76.4% were male (Table 2, Appendix A). The most common etiology of chronic liver disease was hepatitis C virus (*n* = 20, 36.4%). The median treatment duration of immunotherapy was 19 (IQR 10–52) weeks, and 35 (63.6%) patients received combination therapies. Among them, 36 (65.5%) patients had a previous treatment history, while 19 (34.5%) were treatment naïve. The BCLC stages were 4 at stage A, 16 at stage B, 30 at stage C, and 5 at stage D. Following immunotherapy, CR, PR, SD, and PD were observed in 5 (9.1%), 16 (29.1%), 19 (34.6%), and 15 (27.3%) patients, respectively. Overall, 21 patients (38.2%) were classified as responders and 34 (61.8%) as non-responders.

### 3.2. Radiomic Predictor of Treatment Response

Out of 5000 extracted radiomic features for each liver and tumor, data cleaning yielded approximately 3650 and 4225 features as operable for analysis, respectively. On average across the three folds during model development, 12% of liver features and 16% of tumor features were statistically significant (*p* < 0.05), with most being texture-based. The six most predictive features for the liver were: energy, inverse Gaussian left, inverse Gaussian left focus, Gaussian right polar, cluster prominence, and cluster shade. For the tumor, key features included cluster trend, homogeneity, autocorrelation, Gaussian, Gaussian right focus, and Gaussian right polar. Further details are provided in Appendix A. These findings support the hypothesis that distinct CT-based radiomic signatures from both liver and tumor differentiate responders from non-responders. Moreover, no substantial association was observed between the features of the liver and tumor.

### 3.3. Performance of AI-Driven Radiomic Models for the Prediction of Treatment Responses

Predictive performance was evaluated using NB for the liver-only model and five classifiers for the combined model. Three-fold cross-validation was performed, and mean performance across folds was used (Table 3). The accuracy, sensitivity, specificity, and the AUC for the NB model using liver alone are as follows: 0.77, 0.76, 0.78, 0.77. Subsequently, model performance was assessed using both liver and viable tumor features. Improved performance was noted when using both liver and viable HCC features (AUC = 0.81) compared to liver features alone (AUC = 0.77) (Figure 2). The performance metrics—accuracy, sensitivity, specificity, and AUC—for each model were as follows: NB (0.86, 0.71, 0.94, 0.83), KNN (0.79, 0.60, 0.86, 0.84), SVM (0.82, 0.70, 0.86, 0.77), LR (0.76, 0.70, 0.78, 0.74), and DT (0.74, 0.60, 0.78, 0.69). Among them, the NB classifier demonstrated the highest average performance, with superior specificity. These findings suggest that NB may be particularly well-suited for this application due to its simplicity and effectiveness with limited training data. Figure 3 presents the confusion matrices of the NB classifier for Model 2.

## 4. Discussion

A comprehensive analysis of the imaging properties of the whole liver and the tumor was conducted in HCC patients, leading to the development of a model capable of automatically identifying cases that likely to respond to immunotherapy with high precision. The results of the radiomic analysis supported the study hypothesis that the morphology and texture of the whole liver can potentially predict immunotherapy response. When combined with tumor properties, the performance of the proposed classification model was found to be highly satisfactory. The study provided valuable evidence that (1) microlevel characteristics of the whole liver are distinct and exhibit unique feature patterns on CT scans, which are strongly associated with the immunotherapy outcome in HCC patients. (2) The liver’s unique properties, when integrated with tumor properties, ensure an improved prediction of outcome to immunotherapy. (3) AI can efficiently assist in identifying features of both the whole liver and tumor and integrate them to accurately classify CT scans into responders and non-responders categories, prior to initiation of immunotherapy.

Characterizing the liver and viable HCCs as imaging biomarkers is critical for optimizing immunotherapy in HCC. Accurate identification of responders enables personalized strategies that maximize efficacy and minimize ineffective treatment. This is particularly important in the context of advanced HCC, where only 15–30% of patients respond to immune checkpoint inhibitors [19,20]. Even with the survival benefits of atezolizumab plus bevacizumab, a majority of patients fail to achieve durable responses, highlighting the unmet need for reliable predictive biomarkers [21,22]. Prior studies suggest that specific imaging and molecular features stratify patients by treatment benefit. Additionally, uncovering whole liver and tumor heterogeneity and functional properties allows oncologists to customize treatment plans to match individual patient biology, thereby improving survival rates and quality of life for patients with HCC. These findings are also consistent with a recent review that highlighted the role of immune checkpoint inhibitors in unresectable HCC and emphasized the importance of identifying predictive biomarkers to optimize patient selection and outcomes [23]. Our findings align with studies showing that radiomic analysis captures intratumoral heterogeneity and liver background changes that may influence treatment response [12,24]. By integrating this complementary information, our combined model achieved higher overall accuracy and specificity, which could translate to more confident clinical decision-making (e.g., identifying likely non-responders for alternative therapies).

Our study contributes to the expanding field of imaging biomarkers for immunotherapy in HCC. Qi et al. showed that tumor-focused radiomic features, when selected using a genetic algorithm and modeled with XGBoost, could predict short-term immunotherapy response, with a promising performance in a small cohort (*n* = 54) [6]. Qi et al. primarily analyzed the tumor and peritumoral region, whereas our study focused on integrating whole liver features. Xu et al. developed a longitudinal whole liver CT radiomics model, trained using a SVM classifier, and applied SHAP (SHapley Additive exPlanations) to identify and visualize the most predictive features, achieving an AUC of ~0.88 across a multi-institutional dataset (*n* = 395) [24]. This concept aligns with the multi-center study by Vithayathil et al., who used deep-learning to segment the entire liver on CT and constructed radiomic–clinical models using multiple machine learning classifiers and feature-selection methods for patients on atezolizumab–bevacizumab immunotherapy [25]. In their large cohort (*n* = 152), an integrated radiomic–clinical model outperformed the BCLC and Albumin–Bilirubin (ALBI) grade in predicting outcomes (12-month AUC 0.75~0.89). While conventional measures like tumor size, AFP, or ALBI grade offer limited prognostic value, radiomics captures multidimensional liver and tumor characteristics that are not discernible by routine imaging. Compared to these large-scale studies, our model uses handcrafted features with rigorous RFE-based selection and incorporates both liver and tumor features. This approach provides a simple and understandable model structure, making it appropriate for exploratory studies with limited data from a single institution. Furthermore, among the five classifiers (NB, KNN, SVM, LR, DT), NB achieved the best performance, likely due to its robustness in handling small sample sizes and high-dimensional feature spaces that are typical of radiomics-based prediction models. Although the AUC of 0.83 may be considered modest, it is in line with prior radiomics-based models using handcrafted features in limited-size cohorts and highlights the practical challenges in imaging-based immunotherapy prediction.

Notably, our model’s performance (cross-validated AUC of 0.83) is comparable to recent radiomics-based predictors. Yin et al. reported an AUC of 0.85 using a CT radiomics model in 172 HCC patients treated with TACE, programmed cell death protein 1 inhibitors, and tyrosine kinase inhibitors [12]. Similarly, Xu et al. proposed a longitudinal whole liver radiomics model incorporating arterial and portal venous phase CTs before and during immunotherapy, which achieved an external test AUC of 0.875 and was significantly associated with survival outcomes [24]. Despite our single-center design and smaller cohort, our study offers insight into the hepatic microenvironment, which is recognized as a key modulator of immunotherapy efficacy [26]. Chronic liver conditions such as cirrhosis, steatosis, or inflammation alter the tissue architecture and immune microenvironment, potentially leading to texture changes detectable by radiomic analysis [27,28]. These alterations may impact immune cell infiltration, cytokine signaling, or antigen presentation, thereby influencing the effectiveness of immune checkpoint blockade [26].

The primary limitation of this study is the retrospective design and a relatively small sample size, which may increase the risk of model overfitting. The generalizability of our findings remains limited. However, as a proof-of-concept analysis, the dataset sufficiently demonstrates the feasibility and potential of the proposed methodology. Validation in larger cohorts is needed to confirm findings and refine model parameters. We also limited analysis to the portal venous phase to minimize complexity and prevent overfitting. In addition, although we used robust preprocessing (e.g., voxel size resampling and gray-level binning) and RFE-based feature selection to enhance robustness, we did not conduct a separate analysis of the interobserver reproducibility of the radiomic features. Future studies should consider feature-level reproducibility analyses to ensure model robustness. Another constraint is the lack of integration of clinical predictors such as tumor markers due to sample size limitations. Future studies should incorporate clinical and demographic variables to improve prediction robustness. Furthermore, biological interpretation of radiomic features remains challenging and requires larger datasets with matched histopathology to elucidate the underlying biology. Finally, complete automation of the model, including liver and tumor segmentation, was beyond this study’s scope and can be addressed separately.

## 5. Conclusions and Recommendations

In conclusion, this pilot study investigated the distinct radiomic features of the liver and viable HCCs on pre-treatment CT that are associated with immunotherapy response. By integrating both liver and tumor features and applying RFE and multiple machine learning classifiers, we demonstrated that the NB classifier achieved the best performance, with an AUC of 0.83. Our findings suggest that radiomics has potential in capturing imaging biomarkers, particularly in settings with limited data availability. However, the retrospective design and relatively small sample size of this single institution study represent important limitations that may restrict the generalizability of the results. Future research could explore the use of deep learning-based feature extraction or hybrid approaches that combine handcrafted radiomic and convolutional neural network features to enhance model performance. Furthermore, incorporating explainable AI frameworks may improve transparency and clinical trust in model decision-making. Prospective, multi-center validation will also be essential to assess the clinical utility of radiomics-based treatment prediction models in HCC.

## Figures and Tables

**Figure 1 diagnostics-15-02090-f001:**
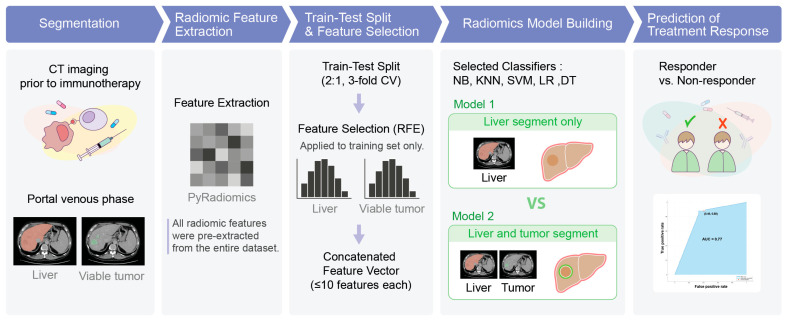
Workflow of the radiomics-based machine learning model for predicting immunotherapy response in HCC patients. Portal venous phase CT scans underwent manual segmentation of the liver and viable tumors. Radiomic features were pre-extracted from all regions of interest using PyRadiomics. In each fold of a three-fold cross-validation, a 2:1 training–testing split was applied. Recursive feature elimination (RFE) was performed independently within the training data for liver and tumor features, and the final selected features were concatenated into a single feature vector. These selected features were then applied to the test set. Although each fold yielded slightly different selected features, a substantial overlap was observed, and commonly selected features across folds were used for model development. Five machine learning classifiers—Naïve Bayes (NB), k-nearest neighbor (KNN), support vector machine (SVM), logistic regression (LR), and decision tree (DT)—were trained to predict treatment response. Final output classified each patient as either a responder or non-responder.

**Figure 2 diagnostics-15-02090-f002:**
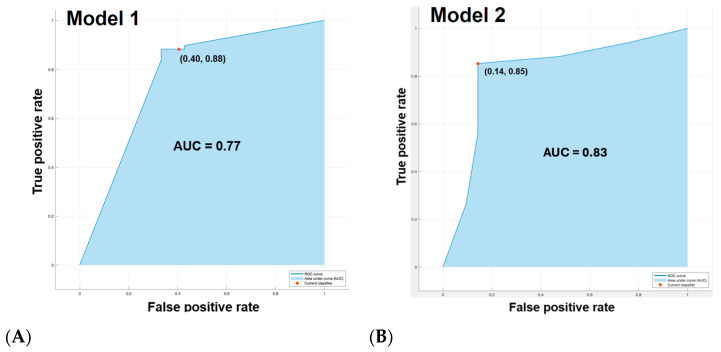
ROC curve comparison of Naïve Bayes classifiers for predicting immunotherapy response in HCC patients. (**A**) Classifier trained using radiomic features from the liver alone (Model 1). (**B**) Classifier trained using combined radiomic features from the liver and viable tumor (Model 2). Each curve represents the average of three-fold cross-validation. NB, Naïve Bayes; HCC, hepatocellular carcinoma.

**Figure 3 diagnostics-15-02090-f003:**
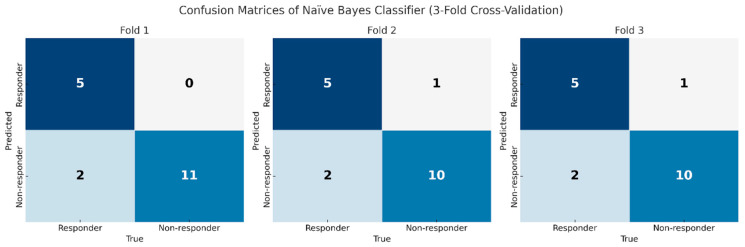
Confusion matrices of the Naïve Bayes classifier for predicting immunotherapy response using combined liver and tumor radiomic features. Each panel displays a confusion matrix corresponding to one of the three cross-validation folds. The model was trained using radiomic features extracted from both the liver parenchyma and viable tumors on CT images prior to immunotherapy. The matrices illustrate the classification of patients into responders and non-responders.

**Table 1 diagnostics-15-02090-t001:** Summary of extracted radiomic features [15].

Feature Type	Feature Examples	No. of Features
First Order Statistics	Kurtosis, Percentiles, Range	15
Gray Level Co-occurrence Matrix	Cluster shade, Contrast, Autocorrelation	20
Gray Level Run Length Matrix	Run percentage, Run entropy	15
Gray Level Size Zone Matrix	Zone percentage, Zone variance	14
Gray Level Dependence Matrix	Small dependence emphasis	12
Shape-based Features 2D and 3D	Volume, Surface area, Sphericity	20
Additional Features	Complexity, Busyness	5

**Table 2 diagnostics-15-02090-t002:** Baseline characteristics of 55 patients.

Variable	Value
Age (years), median (IQR)	69 (63, 73)
Sex, male	42 (76.4)
Etiology of chronic liver disease	
HCV	20 (36.4)
HBV	13 (23.6)
MASLD	9 (16.4)
Alcohol	7 (12.7)
Alcohol + HCV	1 (1.8)
Alcohol + MASLD	1 (1.8)
BCLC stage, A/B/C/D	4/16/30/5
Previous treatment history	36 (65.5)
Immunotherapy type	
Nivolumab	20 (36.4)
Atezolizumab with/without bevacizumab	20 (36.4)
Pembrolizumab	14 (25.5)
Durvalumab + tremelimumab	1 (1.8)
Duration of immunotherapy (weeks), median (IQR)	19 (10, 52)
Treatment response	
Responder (PR + CR)	21 (38.2)
Non-responder (PD + SD)	34 (61.8)

Values are presented as number (percentage) unless otherwise indicated. BCLC, Barcelona Clinic Liver Cancer; CR, complete response; HBV, hepatitis B virus; HCV, hepatitis C virus; IQR, interquartile range; MASLD, metabolic dysfunction-associated steatotic liver disease; PD, progressive disease; PR, partial response; SD, stable disease.

**Table 3 diagnostics-15-02090-t003:** Classification performance of five machine learning algorithms using three-fold cross-validation.

Classifier	Accuracy	Sensitivity	Specificity	AUC
Model 1 (Liver alone)				
Naïve Bayes (NB)	0.77	0.76	0.78	0.77
Model 2 (Liver + viable HCC)				
Naïve Bayes (NB)	0.86	0.71	0.94	0.83
k-NN (KNN)	0.79	0.60	0.86	0.84
SVM	0.82	0.70	0.86	0.77
Logistic Regression (LR)	0.76	0.70	0.78	0.74
Decision Tree (DT)	0.74	0.60	0.78	0.69

AUC, area under the receiver operating characteristic curve; DT, decision tree; KNN, k-nearest neighbor; LR, logistic regression; NB, Naïve Bayes; SVM, support vector machine.

## Data Availability

The data presented in this study are available from the corresponding author on reasonable request. The datasets are not publicly available due to institutional and privacy restrictions, but they can be provided to qualified researchers with the permission of all authors.

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
