# Peer review of "Prediction of Immunotherapy Response in Hepatocellular Carcinoma Patients Using Pretreatment CT Images"

_diagnostics, 2025, doi:10.3390/diagnostics15162090_

Round 1

Reviewer 1 Report

Comments and Suggestions for Authors

I have reviewed your article titled Prediction of Immunotherapy Response in Hepatocellular Carcinoma Patients Using Pretreatment CT Images in detail. In this study, the authors aim to develop a model that performs the process of developing an artificial intelligence model that predicts immunotherapy response through radiomic features obtained from pre-treatment CT images. The study appears interesting in terms of its content and subject matter. However, it appears to have some shortcomings. Providing a few images for each data class in the Dateset section would be helpful for understanding the images studied. It would also be helpful to specify the separation of data for training and testing. The method used for feature selection, shown in Figure 1, should be explained. In the results section, it is important to compare the study with different models to understand its success. However, if available, it should also be compared with other similar studies in the literature. This will allow for better assessment of both scientific robustness and comparative success. Finally, it would be appropriate to ensure that the conclusion section reflects the results of the study under review. By addressing these deficiencies, the scientific basis of the study will be strengthened, the reproducibility of the method will increase, and the contribution of the findings to the literature will be more clearly demonstrated.

Reviewer 2 Report

Comments and Suggestions for Authors

The paper developed an AI-driven prediction model using radiomic analysis from the liver and viable HCCs on pretreatment CT to differentiate responders (those with either partial or complete response to treatment) from non-responders (those who were either with stable or progressive disease even after treatment). The were 2 models: Model 1: binary classification using NB to predict treatment response using only liver features. Model 2: binary classification to predict treatment response using both liver and tumor features. The retrospective study used a small CT dataset that was annotated by 2 experienced radiologists using 3DSlicer and ITK-Snap and exported as DICOM. Radiomic features were extracted using PyRadiomics which complies with IBSI standard. Since there were 5,000 features and not all are relevant to the classification, data cleaning was performed resulting to 3,650 and 4,225 features for liver and tumor respectively. 

The dataset was divided into training (2/3) and testing (1/3) and each ML model was trained using 3-fold cross-validation. RFE was performed to reduced the number of features to a maximum of 10, each for liver and tumor. The best model for Model 2 is NB. For Model 1, only NB was used.

Comments: 

In Fig. 1, there is no information when the train-test split was performed. Was it performed after feature extraction and selection? If so then indicate in the figure. Also RFE was performed but was not indicated in Fig 1. Kindly add RFE to Fig. 1.

In page 4 paragraph 2, you mentioned that 5,000 radiomic features were extracted. Could you provide some explanations how you got that number? 

In page 6 paragraph 1, since RFE was applied and only a maximum 10 features were left, then kindly specify what these features are. I suggest that it be shown in a table to emphasize its importance.

In page 6, paragraph 2, clinical characteristics were reported but I believe this was not used by the models to do binary classification. If so kindly state that the clinical information was presented merely to appreciate the kind of information the dataset contains but such was not actually used in the classification since the models used only radiomic features as can be seen in Fig. 1. 

It is recommended to replace the "Conclusion" section with a "Conclusions and Recommendations" section in order to guide readers how to improve the study and thus move forward. As an example on what to put in the recommendations, deep learning-based feature extraction or a hybrid approach (combination of radiomics-based and CNN) may be pursued. 

Radiomics-based features are hand-crafted featured that includes first-order statistics (intensity features - Mean, variance, skewness, kurtosis, energy, entropy, etc); shape-based features (3D morphology of tumor/liver such as volume, surface area, compactness, sphericity, elongation); texture features (spatial intensity relationships such as GLCM, GLRLM, GLSZM, NGTDM, GLDM); and filtered features (Wavelet, LoG). PyRadiomics extracts handcrafted radiomic features from medical images and their segmented regions of interest (ROI).

Deep Learning methods such as autoencoders or CNN embeddings learn features directly from the image pixels. 

A combination of these 2 methods could potentially improve model performance. 

Another possible item to put in the recommendation is inclusion of an explainable AI component in order to help the clinician understand the logic used by the models in making the classification. Most AI models are not used in the clinical setting due to trust issues with AI. Clinicians need to know what logic was used to come up with the classification. 

There is a 27% similarity according to Turnitin hence a need to rewrite those portions that have large highlights. 

Reviewer 3 Report

Comments and Suggestions for Authors

The topic is interesting and new and the manuscript was nicely prepared. My comments:

1) My main concern about radiological features is the interobserver agreement that sometimes is low. Do you have any data about that?

2) The retrospective design and the relatively limited sample size represent major limitations which limit the applicability of authors' findings

3) The AUC 0.77 seems suboptimal. Any comments?

4) The authors should put their findings in the general context of the literature in the Discussion. In this regard cite the relevant review PMID: 33086471 

Round 2

Reviewer 1 Report

Comments and Suggestions for Authors

I have carefully reviewed the revised version. It is clear that the authors have meticulously revised the work, taking into account the points raised in my previous review, and have expended considerable effort in this process. It is seen that the authors have added the issues such as presenting sample images of the data set mentioned in my previous evaluation, adding comparisons with similar studies in the literature, and arranging the conclusion section to more clearly reflect the findings of the study. These corrections increased the methodological soundness and transparency of the study, strengthened the reproducibility of the method, and made the place and importance of the findings in the literature more evident. In general, I think the study is original in terms of content and an interesting research in the field.

Reviewer 3 Report

Comments and Suggestions for Authors

The revised manuscript is OK. Thank you!